# Parameters Optimization and Test of Caterpillar Self-Propelled Tiger Nut Harvester Hoisting Device

**Xun He, Yanliu Lv, Zhe Qu, Wanzhang Wang \*, Zheng Zhou and Hao He**

College of Mechanical and Electrical Engineering, Henan Agricultural University, Zhengzhou 450002, China; hexun@henau.edu.cn (X.H.); lvyanliu@stu.henau.edu.cn (Y.L.); quzhe071171@henau.edu.cn (Z.Q.); zhouzheng@stu.henau.edu.cn (Z.Z.); hehao@stu.henau.edu.cn (H.H.)
\* Correspondence: wangwz@henau.edu.cn

**Abstract:** Aiming at the problem of a poor separation of tiger nut, soil and grass during harvest, a hoisting device consisting of a combined-type hoisting sieve, vibrating wheels and soil roller was designed in combination with the requirements of the planting and harvesting of tiger nut. Through a theoretical analysis of the movement of the mixture of tiger nut, sand and grass in the process of hoisting, the basic law that affects the soil filter rate was determined, and the parameters affecting the soil-sieving rate were determined, and the hoisting angle, linear hoist speed and range of related parameters of vibrating wheels were obtained. Based on the DEM-MBD method, a simulation model of an excavating and hoisting device was built. With the hoisting angle, linear hoist speed, vibrating frequency and vibrating amplitudes of vibrating wheels as test factors, and the soil-sieving rate as test index, an orthogonal rotating-center combination test with four factors and three levels was carried out. The results showed that the influence of various factors on soil-sieving rate was as follows: vibrating frequency of vibrating wheels > linear hoist velocity > vibrating amplitudes of vibrating wheels > hoisting angle. When the vibrating frequency of the vibrating wheels was 9 Hz, the linear hoist speed was 0.66 m/s, vibrating amplitude of vibrating wheels was 25 mm and hoisting angle was 26°; the maximum value of the soil-sieving rate was 42.5%. The optimized parameters were applied to field test for verification, and the soil-sieving rate of the field test was 44.7%, which was better than the simulation test. The research results can provide a theoretical reference for design optimization and simulation analysis of tiger nut harvesters.

**Keywords:** tiger nut; hoisting device; DEM-MBD; soil-sieving rate

## 1. Introduction

*Cyperus esculentus* (yellow nutsedge, chufa and tiger nut), is also called gingernut and underground walnut. It is a plant with broad utilization and multipurpose uses including as oil, food and feed [1–3]. As its tuber contains rich nutrients including oil, sugar and vitamins, *Cyperus esculentus* is regarded as a promising oil crop, which has been planted in northeast China, northwest China and central China [4–7]. Although *Cyperus esculentus* has small-sized grains, high production and easy planting, ease of management and strong adaptability, it poses a challenge in separating tiger nuts, soil and grass during harvest. This requires a huge amount of manpower and leads to low mechanized harvest efficiency.

Hoisting device is a key component in a tiger nut harvester and their soil-sieving effect exerts large influences on the efficiency in separating tiger nuts, soil and grass during harvest. Tiger nut (Chufa) harvesters that were first developed by western countries were mostly electric haulage drum-sieving types, with rod-guide-type belts mostly adopted in hoisting devices [8,9], which were prone to a high loss rate and poor separation efficiency because of the small particle size of tiger nuts. On the basis of middle- and small-scale tiger nut harvesters abroad, China has successively developed different types of hoisting devices. Jilin Haoyi Harvest Agriculture and Animal Husbandry Science and Technology Development Co., Ltd., Changchun, China) developed conveyor belt scrapers [10]. This hoisting

device is made of steel wire winding, so that its strength is not enough if the feeding quantity is too large, which increases the failure rate. Shandong Juming Machinery Co., Ltd., Zibo, China) developed scraper belts [11]. This device has good transportation performance but poor soil crushing impact and low screening efficiency because of the small effective screening area. Xu Yongjie developed chain-plate segregators [12]. The sieve plate is a strip sieve plate and a baffle is installed on the sieve plate to transport and screen the tiger nut, soil and grass blend. Although this conveyor device can increase the screening area, there is no impact crushing component on the hoisting device, which is not applicable to viscous soil. At present, a few of the existing tiger nut harvesters are improved and designed on the basis of the tuber harvesters, the harvest efficiency and soil-sieving rate of hoisting device remain to be verified and the mechanism of the key components warrants further elucidation [13,14].

Hence, this research developed a caterpillar self-propelled tiger nut harvester (CSTH) and its combined hoisting device by operating with Xinxiang Dilong Pharmaceutical Machinery, Co., Ltd., Xinxiang, China) according to the planting model of *Cyperus esculentus*. The motions of the mixture of *Cyperus esculentus*, soil and grass that are conveyed using the hoisting device were analyzed to determine key factors influencing soil-sieving performance. Based on a multi-body dynamics and discrete element method (MBD-DEM), a simulation model of the hoisting device in a tiger nut harvester was established to obtain the optimal working parameter combinations of the hoisting device by combining with field tests. The obtained results may provide a theoretical reference for improving the working performance and optimizing the mechanical structures of the tiger nut harvester.

## 2. Materials and Methods

### 2.1. Overall Structure and Working Principle of CSTH

The CSTH developed in this research comprises an excavation device, a hoisting device, a threshing and screening device, a scraper type hoisting collection device, an aggregate box, an engine, a hydraulic caterpillar chassis and a wheelhouse, as shown in Figure 1. The CSTH is able to realize the excavation, hoisting and conveying, separation of tiger nut, soil and grass, screening and tiger nut collection through a one-time operation. The technical parameters of the machine are shown in Table 1.

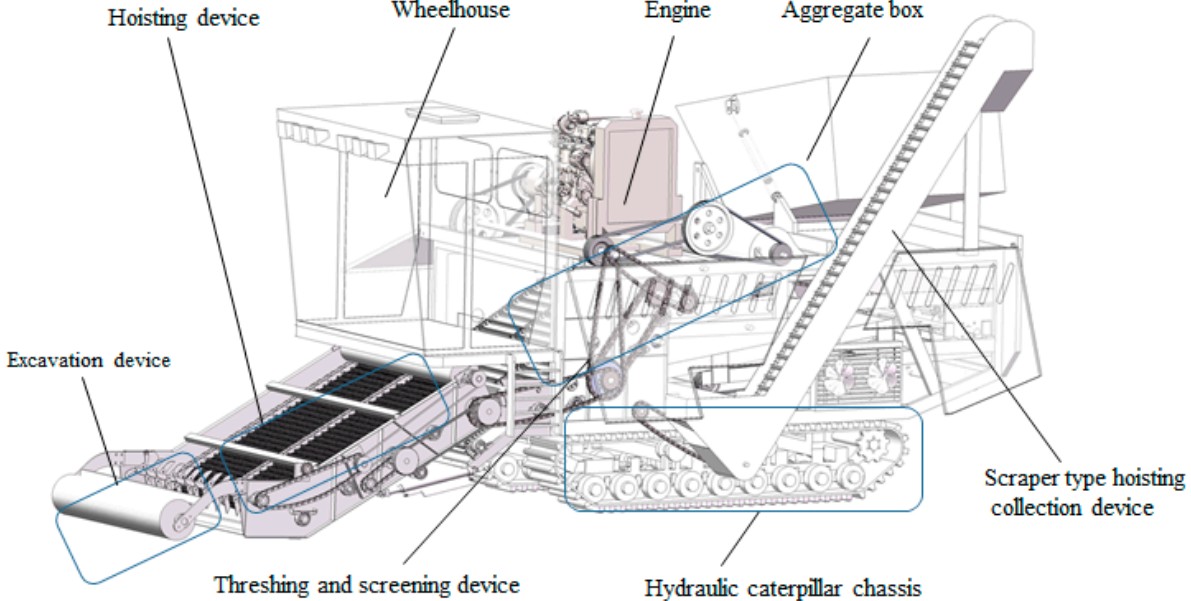

**Figure 1.** Overall structure of CSTH.

**Table 1.** Parameter table of CSTH.

| Item | Value |
| --- | --- |
| Working width/mm | 1800 |
| Excavation depth/mm | 100~200 |
| Overall dimension/mm | $6800 \times 2500 \times 3200$ |
| Auxiliary power/kW | 140 |
| Theoretical production efficiency/hm$^2$/h | 0.133~0.212 |
| Harvesting rate | $\geq$95% |

### 2.2. Working Principle of CSTH

The CSTH during harvest uses a digging shovel to dig soil and utilizes a front-end height-liming wheel to control its operation depth. The blend of tiger nut, soil and grass that are dug out was crushed for the first time using a soil pulverizing cutter and then thrown onto the screening surface of the hoisting device. Afterward, the crushed blend was re-crushed under the combined effects of vibratory sieving and screening of the hoisting device and soil compaction by a soil roller and some part of the soil was filtered out. Furthermore, the remnant aggregates are hoisted and screened, and conveyed to a threshing device. Subjected to the effects of striking, rubbing and squeezing by the threshing device, the tiger nuts were separated from the rhizome of *Cyperus esculentus*. Soil lumps were further crushed into smaller sized soil particles; afterward, the separated particles were thrown into the vibratory sieve on the upper part of the CSTH and a great number of soil particles were leaked out via sieve holes under the effect of reciprocating vibrations, which fell onto the plate, shaking off soil on the lower part of the CSTH, and then exported from the CSTH. Then, the negative-pressure fan absorbs and ejects the lightweight impurities containing *Cyperus esculentus* in the tiger nut, soil and grass blend on the vibratory sieve surface on the upper part. Finally, the tiger nuts obtained were rinsed and then conveyed to the aggregate box by means of the scraper-type-based hoisting collection device.

### 2.3. Design of Hoisting Device

A hoisting device acts as the intermediate link for conveying aggregates from a digging device to the threshing and sieving device. The design of the key component and operation parameters for a hoisting device can determine the working efficiency of a tiger nut harvester. The hoisting device comprises a hoisting chain, vibratory sieving plates, vibrating wheels and soil roller.

2.3.1. Structural Form of Hoisting Chain Structure

At present, most tuber crop harvesters use a rod-type hoisting chain structure. As they are easy to design, show good stability and demonstrate a favorable conveying and separation capability even in the case of high soil moisture content, they have been used in the conveying, crushing and separation of aggregates that are dug out [15]. Aiming at tiger nuts with small tuber sizes, it is necessary to improve the structural design using a rod-type hoisting chain to make the hoisting chain structure meet the technical requirements of hoisting and sieving of the tiger nuts dug out. The vibratory sieving plates were stalled interlaced in the rods of the rod-type hoisting chain to form a combined rod-type hoisting sieve. Among the rods, the hoisting rod was fixed at the hoisting chain according to a given spacing, and then some interlaced-arranged vibratory sieving plates on the rod can form a sieving hole with a certain dimension, which can reduce the harvest loss from the conveying.

During the operation, the tiger nut, soil and grass blend was thrown onto the hoisting sieve surface by a digging shovel and a soil pulverizing cutter. Under the combined effect of the combined rod-type hoisting sieve and vibrating wheels, some particles from the crushed soil lumps were sieved via a sieve hole. Meanwhile, the blend of tiger nut, *Cyperus esculentus* and large soil lumps continued to be conveyed at the rear part of CSTH and further crushed and separated on the threshing and sieving device. The combined

rod-type hoisting sieve had a 7 mm wide sieving hole, the bars were made of steel 10 mm in diameter and the hoisting chain pitch was 63.5 mm, the length was 1620 mm and the width was 1800 mm. The combined rod-type hoisting sieve is shown in Figure 2.

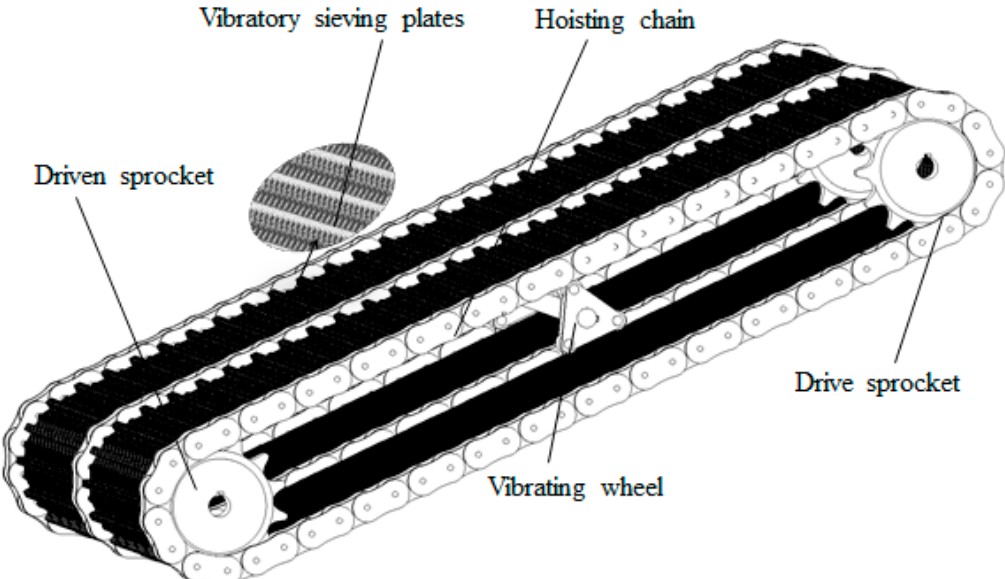

**Figure 2.** Combined rod-type hoisting sieve.

### 2.3.2. Design of Vibratory Sieving Plates

In this research, the vibratory sieving plates on the combined rod-type hoisting sieve are mainly used to be interlaced-installed to form a sieving hole on the sieve surface to avoid the loss of the tiger nuts by leakage from the sieve. The structure of the vibratory sieving plates is illustrated in Figure 3. The 5 mm thick vibratory sieving plates were interlaced in connection to form a 7 mm wide sieving hole. This can prevent tiger nuts from falling from the sieve, which may bring about increasing harvesting loss of tiger nuts; the 16 mm high protrusion on the sieving plates conveys aggregates upward with the rotation of the hoisting chain; there is one ∅6 mm through-hole connecting rod to fix the interlaced-arranged sieving plates on the upper part, while on the lower part, there are two ∅10 mm through-holes sheathed on the hoisting rods, which rotate with the rotation of the drive sprocket to further spur the rotation of the vibratory sieving plates.

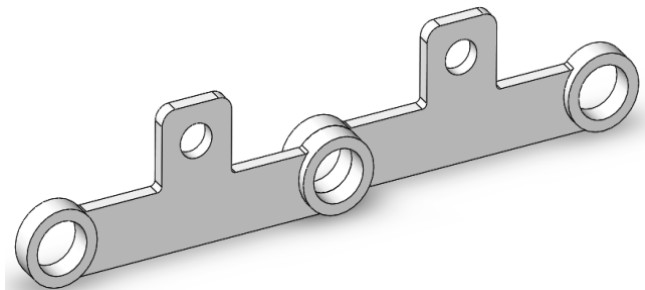

**Figure 3.** Schematic diagram of vibratory sieving plates.

### 2.3.3. Structural Design of Vibrating Wheels

Vibrating wheels are used to make the tiger nut, soil and grass blend on the hoisting sieve surface and to vibrate upward and downward so as to enhance the effects on soil crushing and aggregate separation. The vibrating wheels can be divided into driving-type and driven-type vibrating wheels according to their drive modes. The driving-type shaker can freely adjust the regulation frequency so as to change the frequency of the hoisting chain; the driven-type shaker can produce vibration driven by the hoisting chain, which

matches the linear hoist speed, while the vibration frequency is decided by the hoisting chain [16].

Given that driving-type vibrating wheels need to provide driving force alone, the transmission relationship of the whole machine thus becomes complicated, a driven-type triangle-shaped vibrating wheel with good stability and separation efficiency was adopted. The wheel rotation at each circle under the driving force of the hoisting chain can result in thee reciprocating vibrations [17]. The vibrating wheel was installed at the middle part of the hoisting sieve to vibrate the tiger nut, soil and grass blend conveyed by the hoisting device. Then the soil lumps were crushed to reduce the pileup of aggregates, so as to improve conveying effect of the hoisting device. The schematic diagram of triangular vibrating wheel structure is shown in Figure 4.

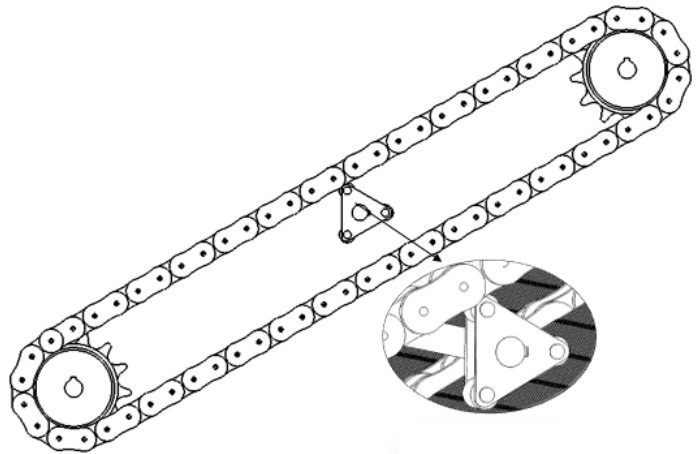

**Figure 4.** Schematic diagram of triangular vibrating wheel structure.

The primary factors of the working performance of the vibrating wheels comprise cam numbers, vibration frequency, vibration amplitude and rotational speed. The dynamic transmission from the vibrating wheels and first-level driving-type wheel is belt-driven, having a transmission gear ratio of 1.11, and the rotational speed of the vibrating wheel is $n_2 = 1.11n_1$. In the meanwhile, according to the relationship between vibrating wheels and the linear speed of the hoisting chains, we obtain:

$$n_2 = \frac{60v'}{2\pi r} \tag{1}$$

The amplitude of the triangle-shaped vibrating wheel commonly used in sandy soil is 10~40 mm [15]; the vibration frequency is derived according to following equation:

$$f = \frac{zn_1}{60} \tag{2}$$

where $v'$ is the linear velocity of the hoisting chain, m/s: $r$ is the radius of the vibrating wheels, mm; $f$ is the vibration frequency, Hz, and $z$ is the number of vibrating wheels, $z = 3$.

### 2.3.4. Design of a Soil Roller

A soil roller was set at the end of the hoisting device to further crush the remained large-sized soil lumps that were sieved and separated in the last step [18]. The soil pulverizing motions are shown in Figure 5. The single *Cyperus esculentus* plant and soil aggregates were compacted under the soil roller to meet the crushing requirements as:

$$f_1 = \mu_1 N_1 \geq P_1 \sin \delta - \mu_2 P_1 \sin \delta \tag{3}$$

$$\sin \delta = \frac{\sqrt{(R_1 + R_2)^2 - (R_1 + h - R_2)^2}}{R_1 + R_2} \tag{4}$$

$$\cos \delta = \frac{R_1 + h - R_2}{R_1 + R_2} \tag{5}$$

Then:

$$R_1 \geq \frac{(2R_2 - h)(1 - \mu_1\mu_2)\left[\sqrt{(\mu_1 + \mu_2)^2 + (1 - \mu_1\mu_2)^2} + 1 - \mu_1\mu_2\right]}{(\mu_1 + \mu_2)^2} + R_2 - h \tag{6}$$

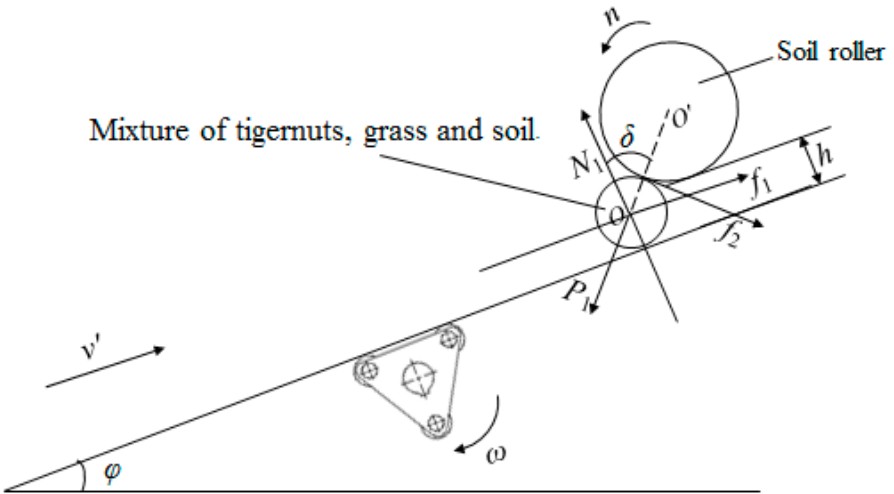

**Figure 5.** Schematic diagram of soil fragmentation.

Based on Equation (6), the radius of the soil roller and the radius of the tiger nut, soil and grass blend, the coefficient of the frictions among the hoisting chain, the soil roller and the blend are set, which are related to the spacing between the soil roller and the sieve surface. Among them, the spacing between soil roller and sieve surface $h$ should be equal to or greater than $R_2$min. Pacing lower than $R_2$min will result in the clogging of soil lumps on the sieve surface, which makes the operation fail to produce reflux, further influencing the conveying efficiency of the hoisting device. On-site measurement was conducted on the radius of the blend with $R_2$ ranging from 50 to 150 mm. Based on the aforementioned analysis, the minimum radius $R_1$min of the soil roller is 53 mm based on Equation (6). The too-large radius of the soil roller will increase power consumption and the weight of the soil roller; however, a too-small radius of the soil roller is likely to cause insufficient crushing of soil aggregates. Under the conditions of meeting conveying demands, the radius and length of the soil roller were set to 100 mm and 1640 mm, respectively, which are in compliance with the width of the hoisting device.

2.3.5. Motion Parameter Setting of the Combined-Type Hoisting Sieve

The factors influencing soil-sieving rate and bean-wounding rate are the motion parameters of vibrating wheels and the liner hoist speed of the hoisting device. The higher the liner hoist speed, the greater the speed at which aggregates are conveyed toward the back. This makes the tiger nut, soil and grass blend fail to be sieved and screened, finally leading to a low soil-sieving rate; when the linear hoist speed is too low, clogging occurs at the front end of the sieving surface; as a result, the hoisting efficiency is reduced. The motion parameters of vibrating wheels mainly comprise rotational speed, frequency and amplitude. When the rotational speed of the vibrating wheels is too fast, the number of collisions between the tiger nut, soil and grass blend and vibrating sieve increases. Although the soil-sieving rate is increased, the collision between tiger nuts and the vibrating sieve leads to increasing the wounding rate of tiger nuts. However, when the vibration amplitude or the frequency of the vibrating wheels is too small, the performance of the hoisting sieving in separating tiger nuts, soil and grass is unsatisfactory,

resulting in a low soil-sieving rate; however, in the case of a too-large vibration amplitude and frequency, the sieve surface will exert large impacts on the blend or throw the blend too high, causing a high wounding rate of tiger nuts and also influencing the conveying, sieving and separating capabilities of the hoisting device. Hence, vibrating wheels should be set with an appropriate vibration frequency, amplitude and rotational speed to increase the separating performance of the tiger nut, soil and grass blend.

The tiger nut, soil and grass blend from the digging device is thrown onto the hoisting device, which is found to be randomly disordered. To ensure the blend is successively conveyed to the hoisting device, the blend subjected to the effect of the vibrating wheels should be thrown while not sliding down without reflux. The throw-conveying process during the hoisting process was analyzed. When the tiger nut, soil and grass blend is subjected to the effect of the triangle-shaped vibrating wheels it performs circular motions. If the weight of the blend is $m$, the angle between the hoisting device and horizontal surface is $\varphi$ and the rotational angle is $\gamma$. The forces borne by the blend include friction force, supporting force, centrifugal force and gravity, as shown in Figure 6.

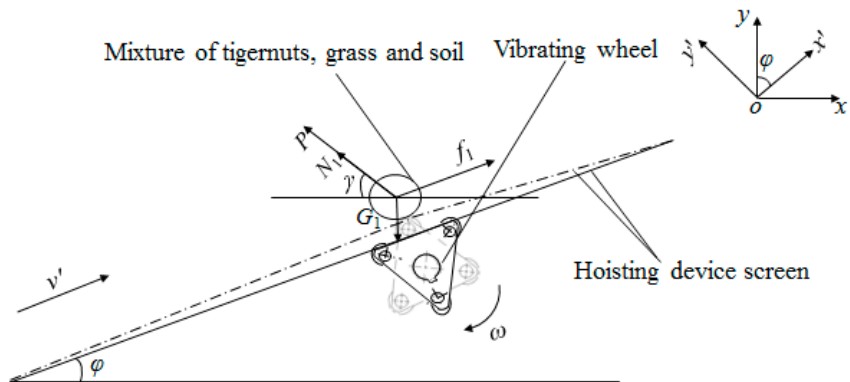

**Figure 6.** Force analysis of mixture of tiger nut, soil and grass blend on hoisting sieve.

Decomposing the forces on the $x$ and $y$ directions gives:

$$P + N_1 = G_1 \sin(\gamma - \phi) \tag{7}$$

$$f_1 = G_1 \cos(\gamma - \phi) \tag{8}$$

Meanwhile:

$$P = m\omega^2 r \tag{9}$$

$$\omega = \frac{2\pi n}{60} \tag{10}$$

$$v = \frac{\omega}{r} \tag{11}$$

To make the tiger nut, soil and grass blend on the hoisting sieve surface be thrown upward, both the supporting force and friction force at the moment when the blend is thrown upward are zero; then we obtain:

$$P \geq G_1 \sin(\gamma - \phi) \tag{12}$$

Simultaneous solution:

$$\omega^2 r \geq g \sin(\gamma - \phi) \tag{13}$$

To prevent the blend on the hoisting chain from sliding down, it is necessary to ensure that:

$$f_1 \geq G_1 \cos(\gamma - \phi) \tag{14}$$

Meanwhile:

$$f_1 = \mu_1 N_1 \tag{15}$$

$$\omega^2 r \le \sqrt{\frac{1}{\mu_1^2} + 1} \cdot \sin(\gamma - \phi + \tan^{-1}\mu_1)g \tag{16}$$

$$\sin(\gamma - \phi)g \le \frac{v^2}{r} \le \sqrt{\frac{1}{\mu_1^2} + 1} \cdot \sin(\gamma - \phi + \tan^{-1}\mu_1)g \tag{17}$$

where $\gamma$ is the rotational angle, °; $\varphi$ is the hoisting angle, °; $P$ is centripetal force, N; $\omega$ is the angular velocity, rad; $f_1$ is the friction force, N; $\mu_1$ is the friction factor; $N_1$ is the holding force, N; and $G_1$ is the gravity of the blend, N.

According to the aforementioned analyses, the main factors influencing the hoisting, sieving and screening of the tiger nut, soil and grass blend are the linear hoist speed of the hoisting device, the vibration amplitude and frequency of vibrating wheels and the hoisting angle. The dip angle of commonly used rod-type hoisting chains is taken as 22°~30° [16,19,20]. The coefficient of friction between the hoisting sieve surface and the blend is 0.3~0.5. Based on Equation (19), the linear hoist speed is calculated to be 0.52~1.23 m/s. Considering that the linear speed for the conveying device of tuber crop harvesters is usually set to be 0.8~2.5 times that of harvesting speed [21,22], the linear hoist speed is set to 0.63~0.81 m/s to avoid clogging of the blend occurring at the front end. By doing so, the harvest efficiency and quality of the tiger nut harvester can be guaranteed.

### 2.4. Simulation Experiments of the Hoisting Device

Based on the aforementioned analysis, the main factors influencing the hoisting, sieving and screening of the tiger nut, soil and grass blend are the linear hoist speed and hoisting angle of the hoisting device, as well as the vibration amplitude and frequency of vibrating wheels. Given that the tiger nut, soil and grass blend collides with different components during tiger nut harvest, which poses a challenge to capture its motion states, the EDEM software was used to analyze the motion state of the particles on the hoisting device; the software RecurDyn (FunctionBay, Gyeonggi, Korea) was adopted to realize the complex rigid body motions of the hosting chain [23,24]. Because the blend was conveyed to the hoisting device after being dug out, a simulation model of the excavation hoisting and conveying device was established based on a coupling simulation to reveal the motion law of the blend in the process of digging out and conveying to the hoisting device. Furthermore, the influences of various parameters on the soil-sieving performance were evaluated.

At first, the chain modules in the software RecurDyn were used to generate the hosting chain and driving and driven sprockets, which were equipped with rotational pair constrains; then, the vibratory sieving plates in the hoisting device were grouped and introduced into RecurDyn, followed by adding the restraints of fixed pairs and rotational pairs; last, the drive was added by means of the driving sprocket to enable the vibratory sieving plates to rotate with the hoisting chain, consequently realizing the uniform and sustainable conveyance of aggregates. Moreover, the EDEM interface in the External SPI module of RecurDyn was adopted to export the vibratory sieving plates as a wall file. Furthermore, using the import function in EDEM, the exported vibratory sieving plates could be imported as a wall file. Meanwhile, the excavation device was imported in EDEM in the STL format to construct the discrete element model for the tiger nuts and soil particles by introducing the advancing motions of a digging shovel and the rotational and advancing motions of a soil-pulverizing cutter. According to the steps above, the coupling simulation model of the excavation, hoisting and conveying device could be built.

In the simulation experiments for simulation analysis, the components that come into little contact with aggregates during the digging, hoisting and conveying process are neglected. Due to complex rigid-body motion-related collisions among the vibratory sieving plates and chains, the function driving the rotation of the hoisting device was only

added, while the feeding of soil and tiger nut aggregates was realized at the speed same as the harvesting speed in an opposite direction set via EDEM.

### 2.4.1. Establishment of Simulation Model

Sandy loam soil has fine-sized particles and is likely to be crushed. As the software EDEM fails to meet the precision requirements of calculating fine-sized particles and shows limitations of computer conditions, the simulation model fails to be built according to the actual particle sizes of the soil. Usually, the images of soil particles will be subjected to enlargement processing. Under the premise of not influencing the accuracy of the simulation model, the radius of soil particles is set to 3 mm, the contact model between soil particles is Hertz–Mindlin with bonding. Soil particles are randomly generated following a normal distribution to meet the requirements in the actual soil-bin soil formation process. Considering that tiger nuts are elliptic particles, the spherical-soil particle aggregation model can be constructed based on the Hertz–Mindlin (no-slip) contact model and the triaxial sizes of tiger nuts, particles with size range of 0.8~1.5 times, are randomly generated according to the actual size range. The simulation model of soil and tiger nut particles is shown in Figure 7.

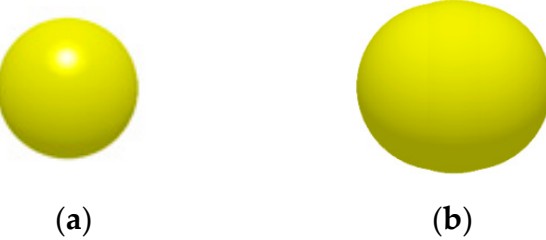

(a)          (b)

**Figure 7.** Simulation model of soil and tiger nut particle: (**a**) Soil particle; and (**b**) tiger nut particle.

The parameters of the simulation model consist of intrinsic parameters and contact parameters of the aggregates. The intrinsic parameters of the aggregates could be determined via bench-scale testing, while contact parameters were calibrated by combining bench-scale testing and simulation experiments to further improve the reliability of discrete element simulation experiments [25]. The parameters for the contact between tiger nuts and steel plates, and the contact between tiger nuts, could be determined by calibration; other parameters are determined by referring to literature [26–30]. Q235 steel was used as geometry material. The simulation test parameters of tiger nuts are shown in Table 2.

**Table 2.** Simulation test parameters of tiger nuts.

| Material | Parameters | Value |
|---|---|---|
| Tiger nut | Density/kg·m$^{-3}$ | 1230 |
| | Poisson ratio | 0.18 |
| | Young's modulus/MPa | 9.44 |
| Steel | Density/kg·m$^{-3}$ | 7850 |
| | Poisson ratio | 0.29 |
| | Young's modulus/MPa | 207 |
| Soil | Density/kg·m$^{-3}$ | 1780 |
| | Poisson ratio | 0.26 |
| | Young's modulus/MPa | 2.77 |
| Soil to soil | Coefficient of restitution | 0.14 |
| | Coefficient of static friction | 0.56 |
| | Coefficient of rolling friction | 0.27 |
| Tiger nut to steel | Coefficient of restitution | 0.619 |
| | Coefficient of static friction | 0.254 |
| | Coefficient of rolling friction | 0.072 |

**Table 2.** *Cont.*

| Material | Parameters | Value |
|---|---|---|
| | Coefficient of restitution | 0.49 |
| Tiger nut to soil | Coefficient of static friction | 0.42 |
| | Coefficient of rolling friction | 0.25 |

2.4.2. Simulation Test Process

A 1200 mm (length), 800 mm (width) and 200 mm (height) soil trough model was plotted combined with the model created above and the simulation parameters; during the simulation test, the forward speed of the excavation hoisting device was set at 0.31 m/s, the excavation depth was set at 130 mm and the experimental index was set as the soil-sieving rate, which referred to the ratio of the mass of the soil particles sieved through the combined rod-type hoisting sieve per unit time to the mass of soil particles that should be conveyed to the hoisting device. In addition, considering the convenience of calculation, the parameters in the simulation calculation should not be too large. The Rayleigh time step for aggregate generation was fixed at 20% and the grid unit size was fixed at three times the minimum particle radius. The simulation process is shown in Figure 8.

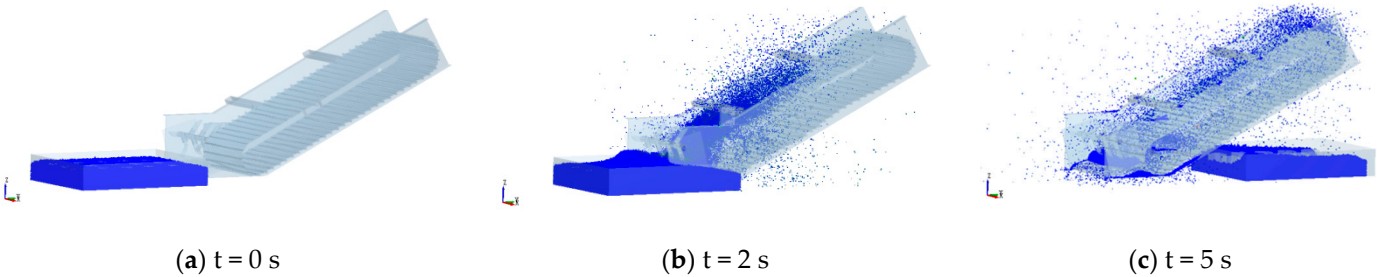

(**a**) t = 0 s        (**b**) t = 2 s        (**c**) t = 5 s

**Figure 8.** Simulation process of excavation, hoisting device.

2.4.3. Quadratic Orthogonal Rotation Combination Design Testing

To obtain the optimal working parameters for the hoisting device, quadratic orthogonal rotation combination design testing was performed. Based on the aforementioned theoretical analyses and the results from previous research [16,17,31,32], the hoisting angle and the linear hoist speed of the hoisting device are 24°~28° and 0.63~0.81 m/s, respectively, and the vibrating frequency and amplitude of vibratory wheels are 7~9 Hz and 10~25 mm, respectively. The coding of simulation test factors can be seen in Table 3, and the experiment scheme and results can be seen in Table 4.

**Table 3.** Coding of simulation test factors.

| Code | Experimental Factors | | | |
|---|---|---|---|---|
| | Hoisting Angle/° | Linear Velocity/m·s⁻¹ | Vibrating Frequency/Hz | Vibrating Amplitude/mm |
| −1 | 24 | 0.63 | 7 | 10 |
| 0 | 26 | 0.72 | 8 | 17.5 |
| 1 | 28 | 0.81 | 9 | 25 |

**Table 4.** Experiment scheme and results.

| No. | Experimental Factors | | | | Experimental Index |
|---|---|---|---|---|---|
| | Hoisting Angle/° | Linear Velocity/m·s⁻¹ | Vibrating Frequency/Hz | Vibrating Amplitudes/mm | Soil-Sieving Rate/% |
| 1 | −1 | −1 | 0 | 0 | 38.58 |
| 2 | 1 | −1 | 0 | 0 | 38.94 |
| 3 | −1 | 1 | 0 | 0 | 38.60 |
| 4 | 1 | 1 | 0 | 0 | 39.80 |
| 5 | 0 | 0 | −1 | −1 | 40.22 |

**Table 4.** *Cont.*

| No. | Experimental Factors | | | | Experimental Index |
| --- | --- | --- | --- | --- | --- |
| | Hoisting Angle/° | Linear Velocity/m·s⁻¹ | Vibrating Frequency/Hz | Vibrating Amplitudes/mm | Soil-Sieving Rate/% |
| 6 | 0 | 0 | 1 | −1 | 41.98 |
| 7 | 0 | 0 | −1 | 1 | 41.46 |
| 8 | 0 | 0 | 1 | 1 | 42.50 |
| 9 | −1 | 0 | 0 | −1 | 39.28 |
| 10 | 1 | 0 | 0 | −1 | 39.98 |
| 11 | −1 | 0 | 0 | 1 | 40.70 |
| 12 | 1 | 0 | 0 | 1 | 39.80 |
| 13 | 0 | −1 | −1 | 0 | 39.30 |
| 14 | 0 | 1 | −1 | 0 | 40.36 |
| 15 | 0 | −1 | 1 | 0 | 41.52 |
| 16 | 0 | 1 | 1 | 0 | 42.12 |
| 17 | −1 | 0 | −1 | 0 | 38.96 |
| 18 | 1 | 0 | −1 | 0 | 39.56 |
| 19 | −1 | 0 | 1 | 0 | 40.46 |
| 20 | 1 | 0 | 1 | 0 | 41.30 |
| 21 | 0 | −1 | 0 | −1 | 39.80 |
| 22 | 0 | 1 | 0 | −1 | 41.80 |
| 23 | 0 | −1 | 0 | 1 | 41.06 |
| 24 | 0 | 1 | 0 | 1 | 40.86 |
| 25 | 0 | 0 | 0 | 0 | 41.40 |
| 26 | 0 | 0 | 0 | 0 | 41.11 |
| 27 | 0 | 0 | 0 | 0 | 40.84 |
| 28 | 0 | 0 | 0 | 0 | 41.50 |
| 29 | 0 | 0 | 0 | 0 | 40.49 |

Using the software Design-Expert 8.0(Stat-Ease Inc., Minneapolis, MN, USA), the analysis of variance for the data in Table 4 was carried out; the results can be seen in Table 5. For the table, it is shown that the experimental model exhibits great statistical significance ($p < 0.0001$), indicating the designed test is reasonable and effective; the determination coefficient is $R^2 = 0.9429$, implying that the regression equation has a good degree of fitting. In addition, the hoisting angle ($A$) and the linear hoist speed ($B$), as well as vibrating frequency ($C$) and amplitudes ($D$), all exert significant influences on the soil-sieving rate; the interaction term between hoisting angle and vibrating amplitudes ($AD$) has a significant influence on the soil-sieving rate ($p = 0.0486 < 0.05$), while the interaction term between the linear hoist speed and vibrating amplitudes ($BD$) shows significant influence on the soil-sieving rate ($p = 0.0101 < 0.05$). The surface diagrams for the factor interaction effect on the soil-sieving rate are illustrated in Figures 9 and 10. The tested results were then subjected to multiple fitting regression to finally derive the regression equation of the soil-sieving rate as follows:

$$T = 41.07 + 0.23A + 0.36B + 0.83C + 0.28D + 0.21AB + 0.06AC - 0.4AD - 0.12BC - 0.55BD - 0.18CD - 1.41A^2 - 0.56B^2 + 0.31C^2 + 0.27D^2 \tag{18}$$

**Table 5.** Analysis of variance of regression equations.

| Source of Variance | Sum of Square | Degree of Freedom | Mean Square | F | p |
| --- | --- | --- | --- | --- | --- |
| Model | 31.71895 | 14 | 2.265639 | 16.51358 | <0.0001 |
| A | 0.653333 | 1 | 0.653333 | 4.761954 | 0.0466 |
| B | 1.569633 | 1 | 1.569633 | 11.4406 | 0.0045 |
| C | 8.3667 | 1 | 8.3667 | 60.98241 | <0.0001 |
| D | 0.918533 | 1 | 0.918533 | 6.694919 | 0.0215 |
| AB | 0.1764 | 1 | 0.1764 | 1.285728 | 0.2759 |
| AC | 0.0144 | 1 | 0.0144 | 0.104957 | 0.7507 |
| AD | 0.64 | 1 | 0.64 | 4.664772 | 0.0486 |
| BC | 0.0529 | 1 | 0.0529 | 0.385573 | 0.5446 |
| BD | 1.21 | 1 | 1.21 | 8.819334 | 0.0101 |
| CD | 0.1296 | 1 | 0.1296 | 0.944616 | 0.3476 |

**Table 5.** *Cont.*

| Source of Variance | Sum of Square | Degree of Freedom | Mean Square | F | p |
|---|---|---|---|---|---|
| $A^2$ | 12.92324 | 1 | 12.92324 | 94.19366 | <0.0001 |
| $B^2$ | 2.063325 | 1 | 2.063325 | 15.03897 | 0.0017 |
| $C^2$ | 0.627379 | 1 | 0.627379 | 4.572784 | 0.0506 |
| $D^2$ | 0.485204 | 1 | 0.485204 | 3.536508 | 0.0810 |
| Residual | 1.92078 | 14 | 0.137199 | | |
| Lack of fit | 1.2361 | 10 | 0.12361 | 0.722148 | 0.6929 |
| Error | 0.68468 | 4 | 0.17117 | | |
| Sum | 33.63973 | 28 | | | |

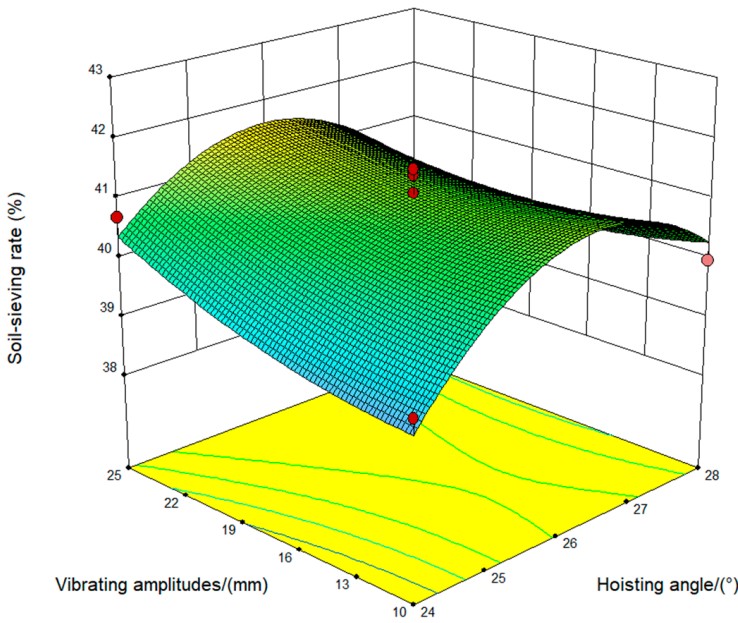

**Figure 9.** Response surface diagram of vibrating amplitudes and hoisting angle.

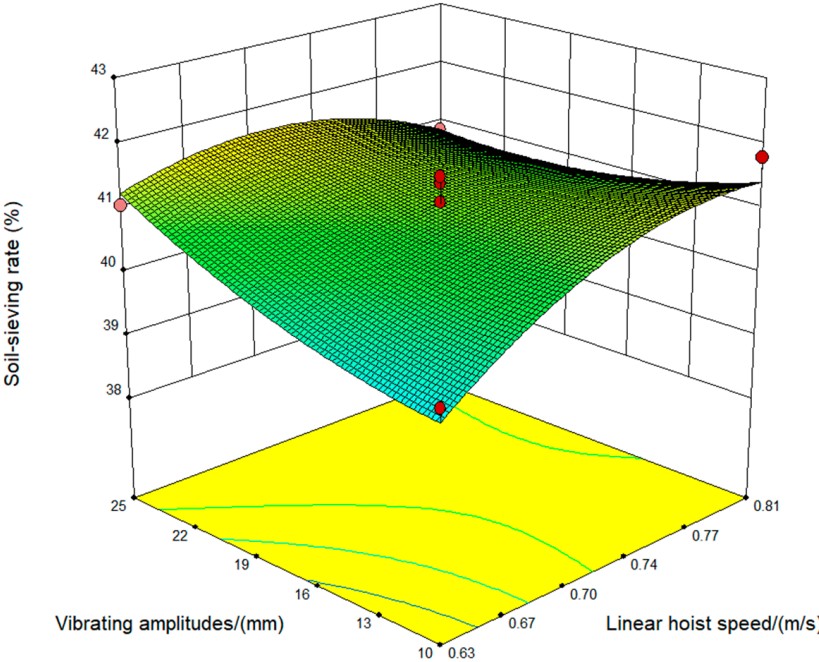

**Figure 10.** Response surface diagram of vibration amplitude and linear hoist velocity.

2.4.4. Optimal Parameters

To obtain optimal parameters, an optimization module was used to solve the regression equation with the aim of maximizing the soil-sieving rate. The objectives and constraints can be expressed in Equation (19):

$$\begin{cases} \max Y(A, B, C, D) \\ \begin{cases} 24 \le A \le 28 \\ 0.63 \le B \le 0.81 \\ 7 \le C \le 9 \\ 10 \le D \le 25 \end{cases} \end{cases} \tag{19}$$

Through the optimizations, the optimal working parameters for achieving the maximal soil-sieving rate of the hoisting device is that the hoisting angle is 26.12° (rounding to 26°), the linear hoist speed is 0.66 m/s, the vibrating frequency is 9 Hz and the vibrating amplitudes is 24.95 mm (rounded to 25 mm).

**3. Field Tests**

A prototype was used to carry out field experiments in Guozhuang village in Xuedian town, Xinzheng city, China, in October 2021, as shown in Figure 11. In compliance with *GB/T 5262-2008 Measuring methods for Agricultural Machinery Testing Conditions—General Rules*, tools including the iSV2101 tachometer, electronic balance, cutting ring, TYY-2 Digital Soil Hardness Tester and YT-SW soil nutrient detector were adopted to measure the field environmental parameters. The variety of the tiger nut is Spanish pellets. Sandy-soil leveling was used as the tiger nut planting mode with plant spacing of about 30 cm, bean-growing depth of 8~13 cm, soil moisture content of 10.5~13%, volumetric weight of 1780 kg/m$^3$ and average firmness of 350 kPa, with a tiger nut moisture content of 44.4%, single grain weight of 0.83 g and volumetric weight of 1179 kg/m$^3$.

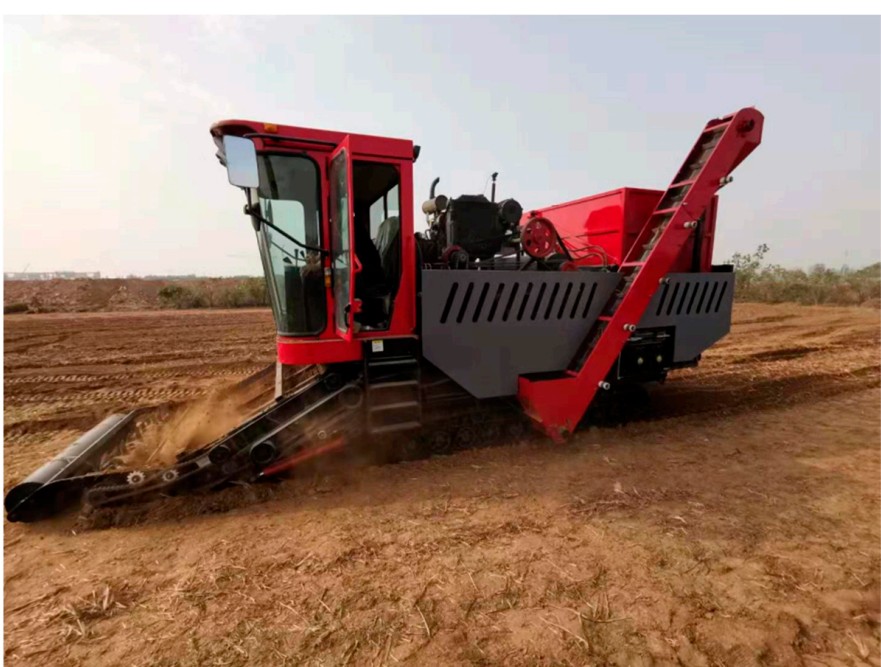

**Figure 11.** Field tests.

*3.1. Assessment Index*

Before a single test, the hoisting working parameters were adjusted to the above-mentioned combination. After rounding, the hoist angle was 26°, the linear hoist speed was 0.66 m/s, the vibrating frequency was 9 Hz and the vibrating amplitude was 25 mm. As there are no designated working standards available at present, the index acquisition

was consistent with Section 3.3, and the soil-sieving rate was taken as the testing index to evaluate the working performance of the hoisting device. The soil-sieving rate refers to the ratio of the mass of the soil particles sieved through the combined rod-type hoisting sieve per unit of time to the mass of soil particles that should be conveyed to the hoisting device. The soil-sieving rate can be expressed in Equation (20).

$$T = \frac{W_1}{W} \times 100\% \tag{20}$$

where $W_1$ is the mass of the soil particles sieved through the combined rod-type hoisting sieve per unit of time and $W$ is the mass of soil particles that should be conveyed to the hoisting device per unit of time.

The digging depth $h$ of the tiger nut harvester under operation is 0.13 m. The length $L$ and width $B$ of the working zone are 2.3 m and 1.8 m, respectively; this zone has a soil volumetric weight of 1780 kg/m$^3$. The zone to be harvested was seen as a rectangle; then the mass of soil particles that should be conveyed to the hoisting device $W$ can be expressed in Equation (21):

$$W = \rho L B h \tag{21}$$

### 3.2. Experimental Methods

The operation of the tiger nut harvester requires a measurement zone with a length equal to or greater than 20 m, a parking zone with a length equal to or greater than 10 m behind the measurement zone and a 5 m long stable zone in front of the measurement zone, the distribution of harvest field test is shown in Figure 12. When the tiger nut harvester was accelerated to a stable speed, a measuring tape installed on the digging device measures a 2.3 m long distance, which is set as the length of the working zone for the CSTH. The harvester stops working at the end of the measurement zone; then the soil on the surface with a height above the digging depth can be weighed, which is the mass of the soil particles sieved through the combined rod-type hoisting sieve $W_1$. The soil-sieving rate refers to the ratio of $W_1$ to $W$.

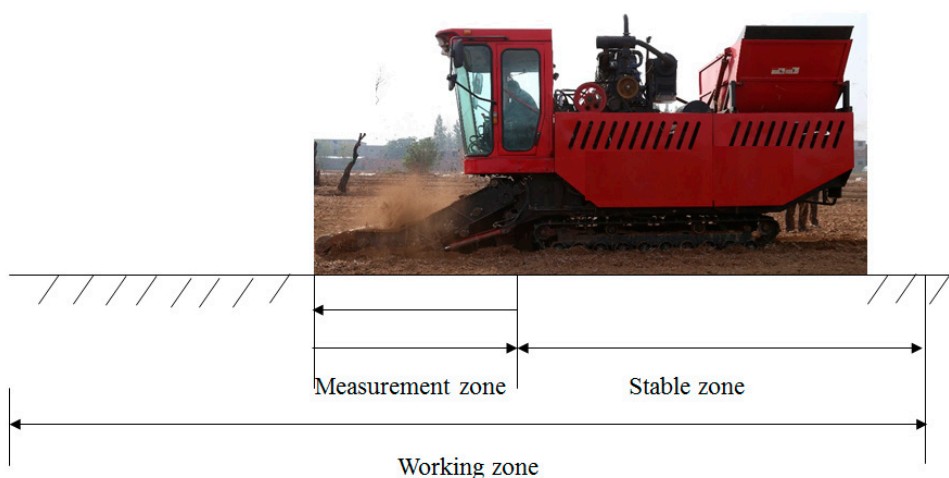

**Figure 12.** Distribution of harvest field test.

### 3.3. Testing Results

Through experimental determinations, it was found that the soil-sieving rate of the hoisting device using the designed optimal working parameter combinations could reach 44.7%, with a loss rate of 1.2% and a damage rate of 1.7%, the measurement results of the field test were shown in Table 6. This finding proved that the device is structurally reasonable and is able to meet the design requirements, largely reducing the working burden of the subsequent cleaning and screening device; consequently, the working efficiency and quality of the CSTH can be improved.

**Table 6.** Measurement results of the field test.

| No | Test Length/m | Soil-Sieving Rate |
|---|---|---|
| 1 | 2.3 | 45.9% |
| 2 | 2.3 | 44.3% |
| 3 | 2.3 | 43.8% |
| 4 | \ | 44.7% |
| 5 | \ | 4.92% |

## 4. Discussion

According to the current situation of the existing tiger nut harvester, the existing problem of the hoisting device is an insufficient screening and crushing of soil and a poor separation of tiger nut, soil and grass. Therefore, a hoisting device composed of hoisting chain, vibratory sieving plates, vibrating wheels and soil roller was designed. Compared with the existing scraper belts, chain-plate segregators and rod guide type belts, the specific performance is that the hoisting device is composed of triangular vibrating wheels and combined-type hoisting sieve, with a simple structure, adjustable screen holes, large effective screening area and better soil crushing and screening performances.

Through theoretical analysis, the key components, such as vibrating wheels, soil roller and combined-type hoisting sieve, were designed, and the structural form of the hoisting device was determined; the key factors and parameter ranges affecting the soil screening effect were determined through the motion analysis of the tiger nut, soil and grass blend on the hoisting screen surface. The motion of the tiger nut, soil and grass blend during the conveying and screening process was simulated and analyzed through the DEM-MBD method. It was clear that when the tiger nut, soil and grass blend was thrown onto the hoisting screen surface, the soil block would roll on the screen surface under the impact force and friction at the moment of contact and then the separation of tiger nut, soil and grass was realized by beating and breaking through the screen. The influence law of various factors on the screening rate and the best combination of working parameters were analyzed. At last, the prototype was trial-produced and the field test was carried out to verify the applicable performance and operation effect of the hoisting device. After the field test, under the optimal combination of operation parameters, the soil-sieving rate of the designed excavation hoisting device could reach 44.7%, indicating that the structure of the hoisting device with the structure of "triangular vibrating wheels and combined-type hoisting sieve" was reasonable and basically met the design requirements, which could greatly reduce the working intensity of the back-end cleaning device and improve the harvesting efficiency and quality to a certain extent. In addition, it can also provide a theoretical reference for the improvement of the working performance and structure of the tiger nut harvester. The simulation and computational ideas mentioned in this paper can also be applied to other crops to solve processing problems similar to those of underground tuber crops.

This study has carried out some innovative structural designs, but there are also some limitations. The specific contents are as follows:

(1) In the simulation study in this paper, the rhizome of the tiger nut is simplified and not considered, and the influence of the presence or absence of the rhizome of the tiger nut on the conveying and screening effect can be considered in the follow-up research.

(2) In addition, the applicability and reliability of the machine in different operating environments should also be considered. The simulation test and field test carried out in this paper were mainly aimed at the sandy loam in Xinzheng city, China. The prototype is still in the pilot stage and has basically met the design requirements; we should study the structural optimization of the hoisting device under different soil types and carry out more field experiments in the further work to improve the operation efficiency and the working capacity of the device.

(3) In China, the mechanized planting of tiger nut is still in its infancy; the designed harvester is only for mechanical harvesting and used in a few places. In the future, intelligent operation technology can be applied to the machine, such as real-time detection of the growth process of tiger nut and real-time identification of harvesting plots and unharvested plots, etc. On the basis of improving the functionality of the harvester, we should further promote the application of the harvester, grasp the advantages of tiger nut as a new multi-purpose crop for oil and grain feeding and actively develop the domestic and foreign markets.

## 5. Conclusions

(1) Based on the agronomic requirement of tiger nut planting and harvest, a hoisting device consisting of hoisting chains, vibratory sieving plates, vibratory wheels, a soil roller, etc. was designed. Its functions include crushing soil lumps, conveying aggregates and separating tiger nuts, soil and grass, etc.

(2) Motion analyses were performed to determine the key components and factors that influence the soil-sieving rate; the coupling simulation model of the digging, hoisting and conveying device was built based on DEM-MBD. Four-factor and three-level quadratic orthogonal rotation combination design testing was conducted by taking the hoist speed, the linear hoist speed and the vibratory frequency and amplitudes of the vibrating wheels as experimental factors, and the soil-sieving rate as an experimental index. The results indicate that the influences of each factor on the soil-sieving rate are shown to be in a decreasing order: the vibratory frequency of vibrating wheels > the linear hoist speed > the vibratory amplitude of vibrating wheels > hoisting angles. The maximal soil-sieving rate reaches 42.5% when the vibratory frequency of vibrating wheels is 9 Hz, the hoist speed is 0.66 m/s, the vibratory amplitude of vibrating wheels is 25 mm and hoisting angle is 26°.

(3) The hoisting device structure with triangle-shaped vibrating wheels and a combined rod-type hoisting sieve and optimized parameter combinations were verified by field tests. The experimental results show that the hoisting device has a maximal soil-sieving rate of 44.7% and can meet design requirements, which provides a theoretical basis for the optimization of a mechanical design for tiger nut harvesters.

(4) Each vibratory sieving plate was made of steel and made by laser cutting; the raised part of the vibratory sieving plate was connected with a rod and the lower end of the vibratory sieving plate was reliably connected to the rod. Therefore, the quality of the parts could be guaranteed, the effective working time of the whole mechanism was long and the failure rate was low. Even if there was a failure, it could be replaced in time alone, which is very convenient for disassembly.

## 6. Patents

Qu, Z.; He, X.; Wang, W.Z.; Zhou, Z.; Lv, Y.L.; Guo, H.Q. Caterpillar Self-propelled Tiger Nut Harvester and Harvesting Method of Tiger Nut: ZL202011123909.8[P]. 2020-12-11.

**Author Contributions:** Conceptualization, W.W. and X.H.; methodology, X.H. and Y.L.; investigation, X.H. and Z.Z.; resources, H.H.; data curation, Z.Q.; writing—original draft preparation, X.H. and Y.L. All authors have read and agreed to the published version of the manuscript.

**Funding:** This research was funded by the special fund for National Key R&D Program of China(Grant No. 2019YFD1002602).

**Institutional Review Board Statement:** Not applicable.

**Informed Consent Statement:** Not applicable.

**Data Availability Statement:** The data used to support the findings of this study are available from the corresponding author upon request.

**Acknowledgments:** The authors would like to thank their college and the laboratory, as well as gratefully appreciate the reviewers who provided helpful suggestions for this manuscript.

**Conflicts of Interest:** The authors declare no conflict of interest.

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
