# Peer review of "Parameters Optimization and Test of Caterpillar Self-Propelled Tiger Nut Harvester Hoisting Device"

_agriculture, doi:10.3390/agriculture12071060_

Round 1

Reviewer 1 Report

Interesting work on the design optimization of a tiger nut harvester.

The main parameters that can influence soil screening are analyzed by simulation and then verified by a real test.

The work may be of interest to improve the design and regulation of these machines, although, given the great limitations that simulation has to reflect the properties of the soil and plants, I believe that it would be important to carry out more real tests to obtain strong results. The authors themselves say that.

The work uses many symbols and formulas, so it is convenient to include a table of symbol equivalences and their meanings.

Throughout the text I have introduced comments on small details that should be improved and also highlighting some detected errors in yellow.

Author Response

Dear reviewer:

       Thank you so much for your time on this paper, thanks for your suggestion. Here is the Response to your comments, please have a look. Thanks again for your work.

Reviewer 2 Report

The authors addressed the design of a tiger nut harvest device. 

- The title of the manuscript is not adequate "Design and Test of Caterpillar Self-propelled Tiger Nut Harvester Hoisting Device". However, the study presented in the manuscript is not explaining the design process, the device is already designed from the beginning of the study. Besides, the device has not been really tested in field conditions, the field test is not explained, there is only one combination of the parameters, and the procedure and results are not clear.

The presented study shows the results of the simulation of a previous designed device to harvest tiger nut. No real field test has been explained.

- A literature review to justify the problem of the poor separation effect of the commercial tiger nut harvesters should be added.

- A review of the working conditions found in previous studies in similar machines (for tiger nut or similar products) should be included (hosting angle, linear hoist speed, vibrating frequency, vibrating amplitude, or similar parameters).

- Lines 58-60: It is not clear if the device was previously designed by the authors, or it is based on a commercial equipment.

- In figures 1 to 4 a picture of the actual equipment should be included (figure 11 is not clear).

- The range of the factors used in the simulation (hosting angle, linear hoist speed, vibrating frequency and vibrating amplitude) should be justify related to previous similar studies.

- In the simulation, it is not clear how the main characteristics of the tiger nut were considered (physical, mechanical, and chemical parameters).

- Was the tiger nut considered as a viscoelastic material?

Which was the shape of the tiger nut used in the simulation? A description of the harvest product is decisive.

- Lines 430-431: The authors explained in the manuscript that the rhizome of the tiger nut was not considered (“In the simulation study in this paper, the rhizome of the tiger nut is simplified and not considered“). In order to harvest a product that is under the ground this part is very important.  Why was it not considered?

- Table 6: caption is not clear.

- Lines 389-397: The explanation of the experimental design of the field test is essential. It seems that the field test was very poor. Only a combination of the experimental factors  was tested (the one that was optimized in the simulation), and the  field experimental conditions are not described.

- Lines 407-408: “Through theoretical analysis, the key components such as vibrating wheels, soil roller and combined-type hoisting sieve were designed”.  A field experiment to test the designed device is imperative.

It is crucial to include a complete field test, with a clear experimental design, a clear procedure, an explanation of the variables measured and an analysis of the results.

The simulation should be validated by extensive experimental results from the field test. It seems that only one combination of the simulation magnitudes was tested. Several combinations of the magnitudes should also be tested in the field test.

It is necessary to explain clearly how the field test was carried out. Some very important parameters about the harvester work are not described (speed of the harvester, percentage of product left, percentage left on the ground, percentage lost in the device, percentage of grass and sand separated…). Also, some variables related to the product (tiger nut) are not included (state of the product, physical and chemical characteristics of the product when harvesting…).

- The conclusions without real field test can not be conclusive.

Author Response

Dear reviewer:

      Thank you for your time on this paper, thank you for your valuable suggestions. I have revised the paper with your suggestion, and learned a lot from your comments. Here is the response to your comments, please have a look.

Reviewer 3 Report

Substantive assessment:
Indeed, the subject of research, i.e. tiger nuts (better known in Europe, especially in Spain as peanut almonds), require specific harvesting machines. Since they are not nuts but roots, they require gentle digging and separation of impurities. Harvesting thousands of tubers the size of a dehusked hazelnut makes the process time-consuming and costly. It is imperative that this collection process be automated, as the authors of this manuscript rightly noticed. Harvesting in rows is a great help. Instead of building a new machine from scratch, it is often worth using solutions that are already known and proven by making only minor modifications. There is a great similarity with the early potato harvest, i.e. the size similar to that of tiger nut tubers. Such a machine is, for example, a potato digger with a conveyor for automatic loading onto means of transport or potato harvesters. These machines are definitely simpler in terms of structure than the presented structure.
The shearer on the caterpillar chassis shown in Figure 1 probably consists of several thousand parts, each of which can damage and immobilize the expensive machine. It is not enough to build a new structure. For the user, its functionality and reliability are of the utmost importance. Especially about the durability and reliability of agricultural machines and vehicles, it is worth including a paragraph, for example, in the Conclusions (which mostly repeat the results already known from previous chapters) or in the previous Discussion.

The proposed harvesting technology is purely mechanical and does not fit into the assumptions of Agriculture 4.0 (in the moment 5.0) - digital farming. This is because it is intelligent agriculture, based on the automation of processes taking into account satellite observations, real-time soil research (based on a soil map) in order to optimize sustainable agriculture. It is worth adapting this technology to the requirements of modern modern agriculture in the future.
The title should contain the Latin name of the subject of research, i.e. the tiger nut.
In order to encourage the use of the developed combine, an economic calculation is needed that will convince potential customers. The machine is dedicated to one plant only, so it will be expensive to operate. It is worth considering selling them to areas where tiger nut is more popular, such as in Egypt, where it comes from.
Should it be clearly indicated, the target, is it intended for humans or perhaps as animal feed?
Why were exactly 29 experiments (table 4) performed, and not, for example, 30 (large random sample in statistics)?

Editorial Rating:
There must be spaces between the numerical value and the unit, e.g. already in the summary (line 22).
What is this unit mu/h (table 1)?

Author Response

Dear reviewer:

       Thank you for your detailed and valuable suggestions, I have revised the paper carefully. Here is the response to your comments from point to point, please have a look. Thanks again for your time.

Round 2

Reviewer 2 Report

The authors have carefully answered all the reviewer´s comments in detail.

The title has been changed to “Parameters Optimization and Test of Caterpillar Self-propelled Tiger Nut Harvester Hoisting Device”.

The literature review has been extended to understand the main parameters involved in the tiger nut harvesting process.

The previous design of the machine has been clearly explained.

A clearer picture of the field machine has been included.

The main work parameters of the tiger nut harvester have been included in section 3.1.

The authors have answered the comments about the field experiment and have clarified the necessary information to understand the purpose and design of the field test.

Suggestions to the authors:

-          A reference indicating the force of the connection between the tiger nut and the rhizome (less than 1N) could be added.

-          The suggestion of a future complete field experiment to confirm the working capacity of the designed device could be included.

Author Response

Thank you again for your comments, thank you so much. I have learned the points of paper writing from all your comments, here are the responses which you can see from the word attachment.
